# Effect of the Porosity, Roughness, Wettability, and Charge of Micro-Arc Coatings on the Efficiency of Doxorubicin Delivery and Suppression of Cancer Cells

**Mariya Borisovna Sedelnikova** [1,*]**, Ekaterina G. Komarova** [1]**, Yurii P. Sharkeev** [1,2]**,**
**Valentina V. Chebodaeva** [1]**, Tatiana V. Tolkacheva** [1]**, Anastasia M. Kondranova** [3]**,**
**Alexander M. Zakharenko** [4] **and Olga V. Bakina** [3]

1   Laboratory of Physics of Nanostructured Biocomposites, Institute of Strength Physics and Materials Science of SB RAS, 634055 Tomsk, Russia; katerina@ispms.ru (E.G.K.); sharkeev@ispms.tsc.ru (Y.P.S.); vtina5@mail.ru (V.V.C.); tolkacheva@ispms.tsc.ru (T.V.T.)
2   Research School of High-Energy Physics, National Research Tomsk Polytechnic University, 634050 Tomsk, Russia
3   Laboratory of Physical Chemistry of Ultrafine Materials, Institute of Strength Physics and Materials Science of SB RAS, 634055 Tomsk, Russia; amk@ispms.tsc.ru (A.M.K.); ovbakina@ispms.tsc.ru (O.V.B.)
4   REC Nanotechnology, School of Engineering Far Eastern Federal University, 690091 Vladivostok, Russia; rarf@yandex.ru
*   Correspondence: smasha5@yandex.ru; Tel.: +7-3822-286-887

**Abstract:** Porous calcium phosphate coatings were formed by the micro-arc oxidation method on the surface of titanium for the loading and controlled release of the anticancer drug doxorubicin. The coatings' morphology and microstructure were examined by scanning electron microscopy. The phase composition was determined with the help of X-ray diffraction analysis. Studies of the hydrophilic properties of the coatings and their zeta potential were carried out. Data on the kinetics of doxorubicin adsorption-desorption were obtained. In addition, the effect of calcium phosphate coatings impregnated with doxorubicin on the viability of the Neuro-2a cell line was revealed. The coating formed at low voltages of 200–250 V contained a greater number of branched communicating pores, and therefore they were able to adsorb a greater amount of doxorubicin. The surface charge also contributes to the process of the adsorption-desorption of doxorubicin, but this effect is not fully understood and further studies are required to identify it.

**Keywords:** calcium phosphate coating; micro-arc oxidation; porosity; doxorubicin; sorption capacity; drugs delivery; tumor cell viability

## 1. Introduction

The development of smart multifunctional biomaterials capable of ensuring drug delivery is the modern trend in medical material science [1,2]. The critical moment of the implant integration into living tissues of an organism is the moment when its surface first binds to human cells and local microorganisms [3]. The modification of an implant surface by various methods is carried out for better biomaterial behavior during the first contact. The surface of the implant may be contaminated with bacteria, which pose a significant threat to the healing process [4]. There is a need to develop systems and materials for the controlled delivery and controlled release of drugs. New drug carriers such as polymer micelles, liposome vesicles, multifunctional dendritic polymers, nanocapsules, nanospheres, and $TiO_2$ nanotubes have been developed with the use of nanotechnology [5]. Many authors propose

the idea of nanotexturing the implant surface and the formation of $TiO_2$ nanotubes, which significantly change the physicochemical properties of the implant surface [4–6].

Plasma electrolytic oxidation (PEO) coatings have an advantage over $TiO_2$ nanotubes [1]. Plasma electrolytic oxidation (PEO), plasma-chemical oxidation (PCO), or micro-arc oxidation (MAO) is one of the most technologically advanced surface modification methods used to create protective, corrosion-resistant, and hardening coatings on the surface of the valve metals. This method can be also used for the synthesis of porous biocoatings [7,8] to modify the surface of orthopedic and dental implants [9]. During the MAO process, the coating is formed as a result of plasma micro-discharges that occur at the coating/electrolyte interface from the components of the substrate and the electrolyte [10]. This leads to the formation of the new crystalline and amorphous phases. The control over the process parameters allows one to vary the coating thickness (up to 100 microns) and its porosity (pore size from submicron to several microns) [1,10]. Porous MAO/PEO coatings are excellent for the impregnation and controlled release of drugs. Diefenbeck et al. [11] reported about PCO coatings on a titanium alloy modified with the help of gentamicin. The coatings combine enhanced osseointegration and antibacterial effects. Peng et al. [12] created a PEO coating on AZ31 with a double Mg–Al hydroxide layer (LDH) by subsequent hydrothermal treatment. The outer layer of Mg–Al LDH on the surface of the PEO coating shows an excellent drug delivery ability.

Bioceramic based on calcium phosphates has clinical recognition in many fields of traumatology, orthopedics, and dentistry. Calcium phosphate (CaP) materials are used in the form of powders, granules, bulk materials; as components of composite materials; and as coatings on metals [13–17]. The interest in calcium phosphate materials as a drug carrier is determined, first of all, by their unique biocompatibility and their properties, which have already been studied. As Son et al. reported [18], brushite bone cement was created using granular β-tricalcium phosphate (β-TCP) and hydroxyapatite (HA) to load with antibiotics gentamicin sulfate solution and release it after setting. Calcium phosphate cement with HA granules demonstrated the prolonged release behavior.

The growth mechanism of CaP coatings during the MAO process is accompanied by the formation of pore spaces in the coatings' structure, which may be a positive factor for the use of porous calcium phosphate coatings as drug carriers. Campos et al. [1] studied the release potential of albumin in four types of CaP PEO coatings in the timeframe of 48 h. It was found that the adhesion and retention of albumin in the coatings is associated with its interaction with Ca and P. It has been found that PEO coatings are suitable for loading and releasing growth factors, antibiotics, and anti-inflammatory drugs.

The antibacterial HA coating induced by polyacrylic acid and gentamicin sulfate (GS) was obtained on a magnesium alloy [19]. The HA coating had antibacterial properties and was capable of prolonging GS release. Thus, PAA/GS-HA coated Mg alloy with good anti-bacterial properties may be an excellent candidate for biomedical applications.

According to GLOBOCAN 2018 reports prepared by the International Agency for Research on Cancer, in 2018 18.1 million new cases of cancer and 9.6 million deaths from cancer were reported [20]. Despite a significant improvement in the situation over the past 10 years, the prognosis for many cancer patients remains unfavorable [21].

Currently, surgical intervention is recognized as the main radical method of treating patients. The local recurrence of the tumor remains the main clinical problem after the surgical treatment of most cancers. Such relapse is usually caused by insufficient resection or implantation during surgery. A promising alternative for the treatment of tissue defects after tumor resection is the use of implantable scaffolds loaded with drugs [21]. Doxorubicin (DOX) is one of the most common chemotherapeutic drugs due to its pronounced antiproliferative and cytotoxic effect [21,22]. The mechanism of anticancer action is based on the ability of the drug to integrate into the DNA of the cell, disrupting DNA repair, and to generate free radicals that cause cancer cell damage [23]. However, DOX has a large number of side effects, which significantly limits its application and prolongs chemotherapy courses [24]. The use of implants containing chemotherapeutic drugs allows one to cancel daily injections and to maintain a constant concentration of the drug, which significantly increases the patient's comfort.

Qiu et al. [25] reported the creation of local implantable scaffolds based on polylactic acid, modified with nanoparticles of mesoporous silica. Such scaffolds were characterized by a high sorption capacity with respect to DOX and the prolonged release of the drug. In the research work [26], lactic-glycolic acid fibers were modified with DOX, which inhibited the cancer by 70% in the experiments in vivo. In addition, a decrease in the side effects of DOX and the improved distribution of the drug in the cancer were observed. A biocompatible polymer matrix based on polyacrylic acid and gelatin loaded with DOX was proposed by the authors as a postoperative implant [27]. In the study [28], nanosized HA synthesized by in situ deposition and coated with DOX and polyvinyl alcohol showed a high cytotoxicity to osteosarcoma cells (MG 63).

The aim of this work was the formation of porous CaP coatings by the MAO method on the surface of titanium for the loading and controlled release of the anticancer drug DOX and research of the influence of the coatings' properties and porous structure on the efficiency of DOX delivery and the suppression of cancer cells.

## 2. Materials and Methods

### 2.1. Sample Preparation

The commercially pure titanium (99.58 Ti, 0.12 O, 0.18 Fe, 0.07 C, 0.04 N, 0.01 H, wt.%) was used for sample preparation. The experimental samples $10 \times 10 \times 1$ mm$^3$ were ground by silicon–carbide sandpaper No. 1200 grit, and then were cleaned ultrasonically in distilled water for 10 min. The average roughness ($R_a$) of the samples was 0.3–0.6 μm. The Micro–Arc 3.0 installation (ISPMS SB RAS, Tomsk, Russia) used to realize the MAO method for the deposition of the coatings was the same as that reported in the previous works [29,30]. Micro–Arc 3.0 includes a pulsed power source, a computer for setting the parameters and the controlling of the coating process, a galvanic cooled bath with electrolyte, and a set of the Mo electrodes. The electrolyte solution containing 30% $H_3PO_4$ consisted of the following components: $CaCO_3$ (50–75 g/L) and stoichiometric HA nano-powder (40–60 g/L). The MAO process was carried out with the use of the anodic potentiostatic regime with the following parameters developed in the previous research [31]: a pulse frequency of 50 Hz, a pulse duration of 100 μs, and a process duration of 10 min. The pulsed voltage varied in the range of 200–300 V.

### 2.2. Experimental Methods

The morphology and microstructure of the coatings were examined by scanning electron microscopy (SEM) with the help of LEO EVO 50 electron microscope (Zeiss, Germany) in the "Nanotech" center at ISPMS SB RAS. The average pore sizes in the coatings were measured by the secant method according to ASTM E1382–9 [32] and DD ENV 1071-5 [33]. The porosity was calculated by the formula: $P$ (%) = $\Sigma l / \Sigma L \times 100$, where L is the full length of the secants on the SEM images and l is the length of the secants within the pores as in the previous research [34]. The number of secants was 50 per specimen. The surface roughness was estimated with a Hommel–Etamic T1000 profilometer (Jenoptik, Jena, Germany) by the average roughness ($R_a$). The traverse length and rate of the measured profile were 6 mm and 0.5 mm/s, respectively.

The phase composition was determined with the help of X-ray diffraction analysis (XRD, DRON–7 "Nanotech" center, Burevestnik, St. Petersburg, Russia) in the angular range of $2\theta = 10°–95°$, with a scan step of 0.02° with Co K$\alpha$ radiation. The interpretation of the XRD-patterns and the phase identification were performed using the powder database of the Joint Committee on Powder Diffraction Standards (JCPDS).

The wettability tests were carried out with the use of the goniometer Easy Drop (Kruss, Hamburg, Germany) with the drop shape analysis software DSA1 (Kruss, Germany), as was exhibited in [31]. The free surface energy of the coatings was calculated in accordance with the Owens–Wendt equation [35]. For the zeta potential analysis of the coatings, the SurPASS™ 3 Electrokinetic Analyzer (Anton Paar, Graz, Austria) was used. For each measurement, the corresponding sample was fixed on

a sample holder (with a cross section of $20 \times 10$ mm$^2$) using double-sided adhesive tape. A 0.001 mol/L KCl solution was used as the background electrolyte, and the pH of this aqueous solution was adjusted with 0.05 mol/L of HCl and 0.05 mol/L NaOH, respectively.

### 2.3. Kinetic Curves of the Sdsorption of DOX by the CaP Coating Samples

For experimental studies, the DOX hydrochloride brand TEVA (Israel) was used. An isotonic solution containing 0.85% NaCl was prepared as a solvent. The DOX concentration in the solution was determined by spectrophotometry at a wavelength of 480 nm; in this case, the optical path length of the cell was 10 mm (spectrophotometer SF-2000, LOMO, St. Petersburg, Russia).

To determine the concentration of DOX, a calibration graph was constructed as follows. An initial solution of DOX (stock solution) at a concentration of 50 µg/mL was prepared. Then, a series of DOX solutions with concentrations of 50, 25, 12.5, 6.3, and 3.13 µg/mL were prepared with the use of the subsequent dilution. The determination of the optical density was carried out for all the prepared solutions. Isotonic solution 0.85% NaCl was used as a comparison solution. Based on the obtained data, a calibration graph was constructed (Figure 1), which was subsequently used to obtain the kinetic adsorption-desorption curves.

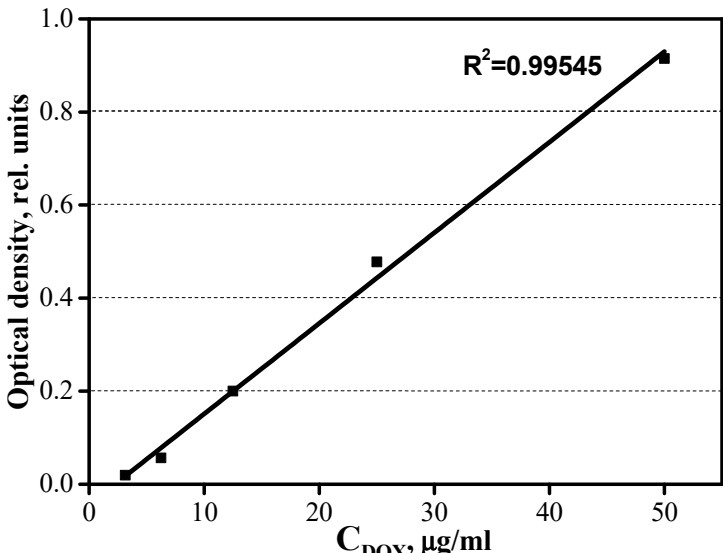

**Figure 1.** Calibration graph of the dependence of doxorubicin DOX concentration on the optical density.

Adsorption kinetics tests were performed at a constant temperature of 25 °C. The CaP coating on a Ti substrate was placed in a glass vessel containing 10 mL of an aqueous solution of DOX at a concentration of 50 µg/mg. The mixture was stirred at a speed of 200 rpm (mechanical shaker Biosan, Riga, Latvia) for a predetermined time. Before determining the concentration of DOX, the solutions were filtered through a 0.45 nm nylon membrane. The concentration of DOX in the solution was determined by spectrophotometry.

The amount of DOX adsorbed for specific periods of time was calculated by the following Equation (1):

$$Q_t = \frac{(C_0 - C_t)V}{m} \qquad (1)$$

where $Q_t$ (mg/g) is the amount of DOX adsorbed to 1 g of adsorbent over time $t$ (min); $C_0$ (mg/L) and $C_t$ (mg/L) are the initial and residual DOX concentration in solution at time $t$, respectively; $V$ (l) is the volume of the solution; $m$ (g) is the mass of the adsorbent.

### 2.4. Kinetic Curves of the Desorption of DOX by the CaP Coating Samples

To study the release of DOX from the modified samples, they were immersed in 10 mL of physiological medium (0.9 wt.% NaCl). at 37 °C. The sampling of physiological medium and the determination of the concentration of the released DOX were performed for 7 days. Further samples' aging immersion led to the destruction of DOX and to significant changes in the absorption spectrum. The samples were placed in a series of flasks containing 10 mL of DOX solutions with a concentration of 50 µg/mL. The adsorption was carried out at the temperature of 25 °C and with constant stirring for 150 min until the adsorption-desorption equilibrium was reached. After adsorption, the samples were rinsed with reverse osmosis water and placed in test tubes containing 5 mL of isotonic solution. The tubes were placed in a thermostat at 37 °C with constant shaking. The samples were taken for 10 days and the DOX content was determined spectrophotometrically.

### 2.5. Study of the Effect of the Samples on Cell Line Viability

The cell culture Neuro-2a was used for cell viability research by the method of cell counting in the Goryaev chamber. This is a mouse neuroblastoma, a clone of the neuro C-1300 cell line, fibroblast-like and with neuron-like cells with nuclei of various shapes. The Neuro-2a cell line was obtained from State Research Center of Virology and Biotechnology "VECTOR", Novosibirsk, Russia. The cells contain large blocks of chromatin, with nucleoli of various shapes and sizes. For the experiment, the cell cultures were dispersed into Petri dishes to obtain a monolayer of cells, which was washed with versene. The cells with 3 mL of trypsin-versene were trypsinized, and 3 mL of culture medium Minimal Essential Medium (MEM) with the addition of 10% fetal bovine serum (HyClone, Marlborough, MA, USA), 2 mM L-glutamine (HyClone, Marlborough, MA, USA), and 1% penicillin/streptomycin (HyClone, Marlborough, MA, USA) was added to each dish.

The resulting cell suspension was centrifuged at 400 rpm for 5 min. Then, the supernatant was discarded and fresh nutrient medium was added with constant stirring by repeated pipetting. Then, 25 mL of the cell suspension was prepared at a concentration of 50,000 cells, and 1 mL was placed in each well of a 24-well plate. The cells were incubated for 24 h at the temperature of 37 °C and $CO_2$ 5% (Sanyo MCO-5AC, Japan). After the incubation, the cells were washed with Dulbecco's sodium phosphate buffer DPBS (HyClone, Marlborough, MA, USA). The samples of CaP coatings on Ti were placed on a cell monolayer. The plates were incubated for 24 h at 37 °C, $CO_2$ 5%. Then, the samples were removed from the wells, the cells were washed, and 0.5 mL of trypsin-like solution was added. Further, the cells were detached, stained with trypan blue, and counted in a Goryaev chamber. As a negative control, the cell viability in the culture medium was examined. As a positive control, the cells with DOX at a concentration of 0.1 µg/mL were used.

To assess the effect of the samples impregnated with DOX on the viability of cancer cells, an MTT assay based on 3-(4,5-dimethylthiazol-2-yl)-2,5-diphenyl tetrazolium bromide was used. The test is based on the study of the inhibition of the mitochondrial respiration of cells. MTT is a yellow salt of 3-[4,5-dimethylthiazol-2-yl]-2,5-diphenyltetrazolium bromide which turns purple when exposed to mitochondrial enzymes. The cells with living mitochondria can restore the reagent, so the color intensity is proportional to the number of undamaged mitochondria [36]. An MTT solution was prepared as follows: 200 mg of the dye was dissolved in 40 mL of DPBS, and a dye concentration of 5 mg/mL was obtained. The solution was used immediately after it was prepared.

To obtain a calibration graph, 150 µL of a cell solution of 500–25000 cells per well and 20 µL of MTT solution were added to the wells of a 96-well plate (TPP, Switzerland). The plate was incubated for 2 h in a $CO_2$ incubator. The growth medium was replaced with dimethyl sulfoxide (99.9%) in each well after the incubation. The plate was placed in a microplate spectrophotometer Thermo Scientific Multiskan FC (Thermo Fisher Scientific, Waltham, MA, USA) and the optical density in each well was determined at 570 nm. An example of a full plate is shown in Figure 2a.

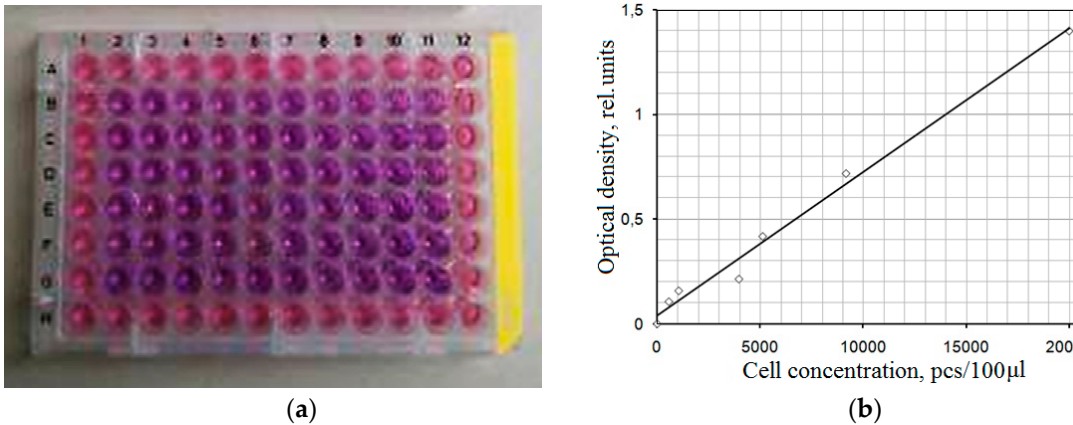

(**a**)                                                                                                    (**b**)

**Figure 2.** Photo of the culture plate before measuring the optical density (**a**); calibration graph of the dependence of the number of cells on the optical density (**b**).

Based on the obtained data, the calibration graph of the dependence of optical density on the number of living cells in the cell suspension (Figure 2b) was plotted.

To carry out the experiments with the CaP coatings on Ti, the samples were pre-impregnated with a DOX solution and were added to a series of flasks containing 10 mL of DOX solution with a concentration of 50 µg/mL. Adsorption was performed at a temperature of 25 °C and with constant stirring for 150 min until the adsorption-desorption equilibrium was reached by all the samples. After adsorption, the samples were rinsed with reverse osmosis water and placed in test tubes containing 5 mL of cultural medium. The tubes were placed in a thermostat at 37 °C with constant shaking. After 24 h, 150 µL of cell medium was taken from the tubes.

A Neuro-2a cell culture with a cell medium containing extracts from samples was incubated at the temperature of $37 \pm 1$ °C and 5% $CO_2$ for 24 h.

For the MTT test, the nutrient medium was removed and the cells were washed twice with DPBS. Then, 100 µL of the culture medium and the 10 µL of 5 mg/mL MTT solution were added to each well of a 96-well plate. Incubation with MTT solution was carried out for 2 h at the temperature of $37 \pm 1$ °C and 5% $CO_2$. At the end of the incubation, the culture medium was removed and 100 µL of dimethyl sulfoxide was added to each well and photometric as described above. Then, the percentage of the living cells was calculated by the ratio of the optical densities of the test and the control samples. The experiment was performed for 10 days.

## 3. Results

### 3.1. Formation of the CaP Coatings and their Structure, Morphology, and Porosity

The CaP coatings were deposited on the surface of titanium plates by the MAO method at different process voltages (Figure 3). The studies showed that the applied voltage dramatically affects the surface morphology, relief, thickness, and roughness of the coatings. It is known that the surface morphology and developed relief play a significant role in the interaction of CaP coatings with biological fluids and living systems (cells, bacteria) [37] and when they are used as carriers of drugs. In this case, the porous structure of the coatings plays a special role, since the nature of the porosity, the size, and the pore shape have a significant effect on the processes of drug adsorption and its controlled release [38].

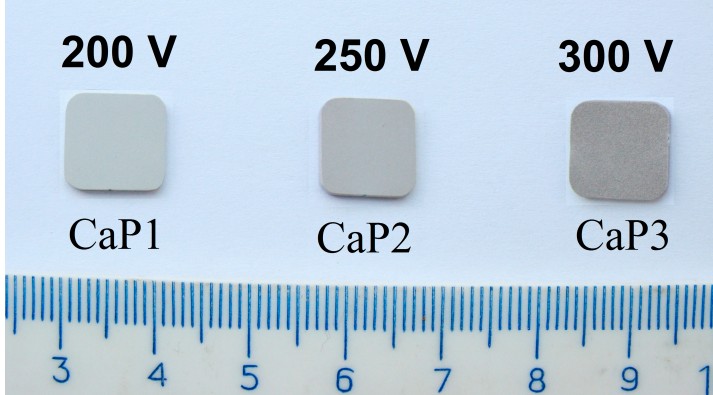

**Figure 3.** Photographs of the calcium phosphate (CaP) coatings synthesized on titanium at different process voltages of the micro-arc oxidation (MAO).

The SEM studies demonstrated that CaP coatings with different surface morphologies were formed depending on the applied voltage (Figure 4a–c). At a pulse voltage of 200–250 V, the formation of spheroidal structural elements (spheres and hemispheres) with open pores and pores located in the recesses between the spheres occurred on the surface of the coatings.

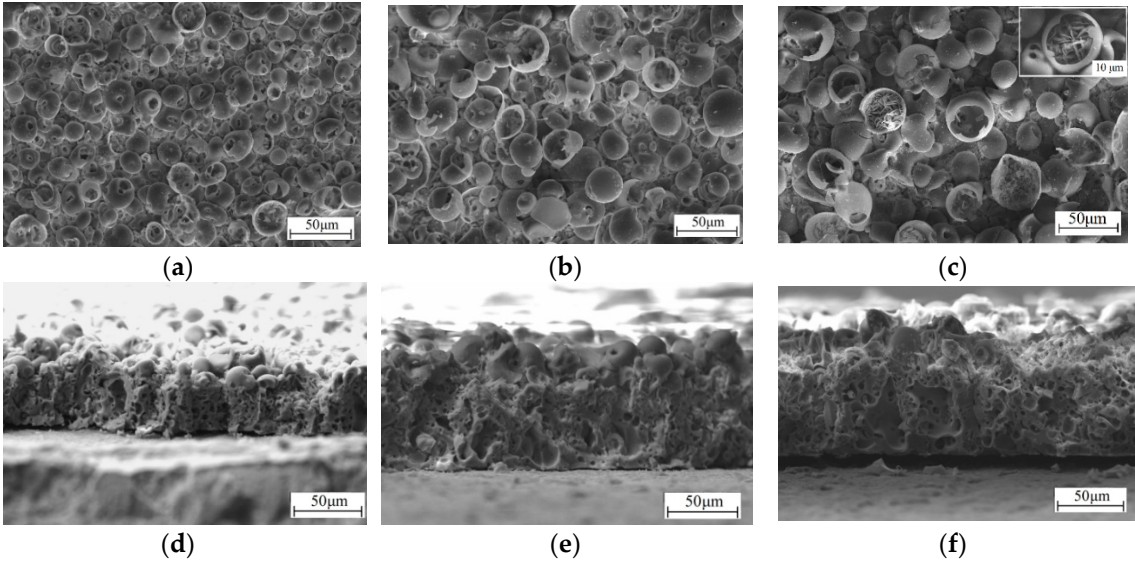

**Figure 4.** SEM images of the surface (**a**–**c**) and cross section (**d**–**f**) of CaP coatings obtained at different process voltages (*V*): CaP1 200 (**a**,**d**), CaP2 250 (**b**,**e**), CaP3 300 (**c**,**f**).

An increase in the applied voltage from 250 to 300 V led to an increase in the intensity of micro-arc discharges acting on the substrate; as a result, the surface morphology of the coatings changed. The sizes of the spheroidal structural elements increased, and they partially collapsed with the formation of fragmentation elements. At the same time, numerous plate-shaped crystals with a length of up to 15 μm were formed on the surface of the destroyed hemispheres (Figure 4c).

A further increase in the pulse voltage to 400 V was accompanied by the destruction of the spheroidal structural elements and the formation of numerous crystals and fragments that completely filled the free pore spaces. At such voltages, the micro-arc discharge transformed into an arc discharge; as a result, the structural elements of the coating were destroyed. Thus, during the formation of CaP coatings on the surface of bioinert titanium by the MAO method, the voltage range of 200–300 V was optimal.

In this voltage range, the total porosity of the coatings increased from 32% to 38% (Figure 5a), while the average pore size increased in the range of 4 to 15 µm, respectively (Figure 5a). In this case, the thickness and roughness of the coatings increased from 50 to 110 µm and from 4 to 7 µm, respectively (Figure 5b).

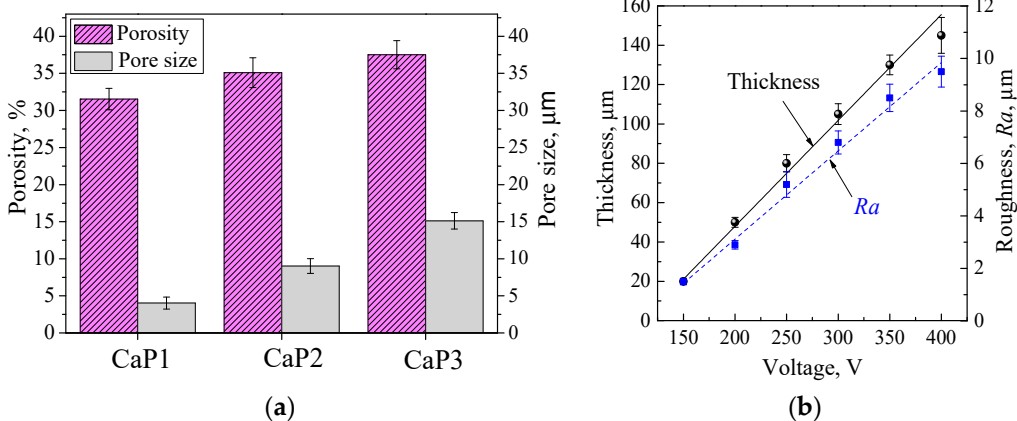

**Figure 5.** Dependences of the porosity and pore sizes (**a**) and thickness and roughness *R*a (**b**) of the CaP coatings on the process voltage.

Spheroidal structural elements were found on the surface of the coatings only, as the SEM images of the CaP coating cross sections demonstrate (Figure 4d–f). A complex porous structure with a large number of branched pore channels uniformly distributed over the thickness is seen inside the coatings. Such porous structure is a characteristic feature of MAO coatings. In the MAO process, a large number of short-lived micro-arc discharges appear as a result of the localized electrical breakdown of the growing coating. These discharges play an important role in the coating growth mechanism and leave characteristic pores in the structure of the growing coating. The sizes of the internal pores also increased, and the formation of single local large pores-containers with sizes of 10–30 µm took place mainly in the coating layers close to the substrate (Figure 4f).

The formation of such pores in the coatings is associated with an increase in the intensity of micro-arc discharges when the applied voltage increases above 250 V. Under the action of a high voltage, intense cascades of micro-arc discharges in a localized region are created in the initial stage of the MAO process, leading to an increase in the size of the internal pore spaces [34].

*3.2. Phase Composition of CaP Coatings*

The phase composition of the coatings should provide their indifference to the drug solution, which is impregnated in the pores of the coatings. The studies by the XRD method showed that the CaP1 MAO coatings formed at the voltage of 200 V were mainly in an X-ray amorphous state, as evidenced by the diffuse halo at the angles ($2\theta = 20°–40°$) (Figure 6). In this case, the reflections of the $\alpha$-Ti (ICDD 44-1294) corresponding to the substrate material were presented on the X-ray diffraction pattern. With an increase in the process voltage, a structural-phase transition from an X-ray amorphous state to an amorphous-crystalline state was observed in the coatings. Reflexes of the crystalline phases such as monetite, $CaHPO_4$ (ICDD 09-0080), $\beta$-calcium pyrophosphate, and $\beta$-$Ca_2P_2O_7$ (ICDD 09-0346) can be seen together with the diffuse halo in the X-ray diffraction patterns of the coatings CaP2 and CaP3, which are formed at the voltages of 250 and 300 V respectively. Their intensity grows when the voltage is increased.

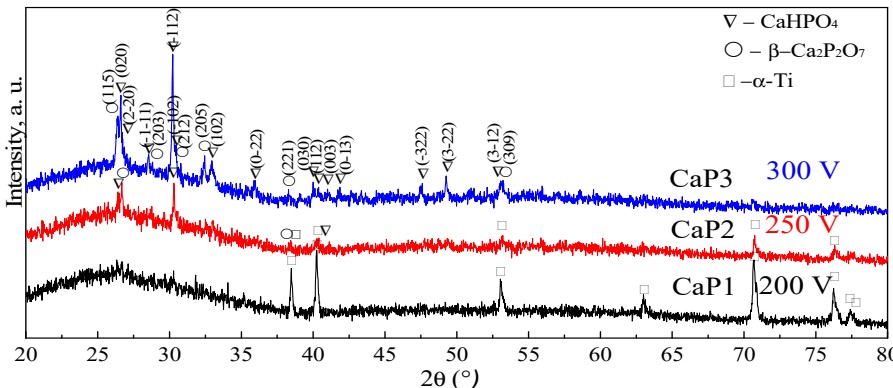

**Figure 6.** XRD patterns of the CaP coatings obtained at different process voltages (*V*).

These data are consistent with the SEM results illustrating the accumulation of plate-shaped crystals in the coatings obtained at high voltages of 250–300 V (Figure 4c). A similar form of crystals is typical for monetite.

The formation of crystalline phases in the coating is a positive fact, since the resistance to the dissolution of $CaHPO_4$ and $\beta\text{-}Ca_2P_2O_7$ is much higher than that of amorphous calcium phosphates [17]. A certain orientation of the crystals is important, since different types of crystallographic planes have different charges. Due to this fact, the selective adsorption of various ions and organic compounds can take place [39].

### 3.3. Hydrophilic Properties, Surface Energy, and Charge of CaP Coatings

It is known that the hydrophilic properties of biomaterials positively affect the adsorption of proteins on the surface of implants and, as a result, enhance the adhesion and proliferation of pre-osteoblasts [40]. It was also shown [41] that hydrophilic surfaces stimulate biomineralization processes. The good wettability of the surface promotes the adsorption of the drug solution by the porous CaP coating and its retention and slow release. Studies of the coatings' wettability showed that the contact angles with water and glycerin did not exceed 25° ± 2°, which indicates the good hydrophilic properties of the coatings (Figure 7a). This may be due to the developed surface topography and porous morphology, as well as the chemical composition of the coatings. The contact angle decreased with the increasing of the applied voltage, especially in experiments with glycerin.

The values of the surface energy were as high as 75 ± 2 mN/m (Figure 7b). Moreover, the polar component significantly predominated over the dispersion component. The electrostatic forces generated by polar groups are responsible for the polar component of the surface energy. Polar groups are generated due to molecular structures that have lost part of their atoms and due to chemical bonds forming electrically neutral "dipoles" [42]. The data indicate the presence of strong polar chemical bonds in the coatings containing $PO_4^{3-}$ and $OH^-$ groups. This is also confirmed by the XRD results (Figure 6), demonstrating the presence of the crystalline phases $CaHPO_4$ and $\beta\text{-}Ca_2P_2O_7$ in the coatings. In addition, the cells' adhesion and proliferation mainly depend on the polar component of energy and significantly increase with the gain in the polar component.

The effectiveness of the interaction of the implant with living tissues is also determined by the electrical properties of its surface [43]. There are examples of numerous studies regarding the induced zeta potential and its role in the engineering of biological tissues on different artificial surfaces [44].

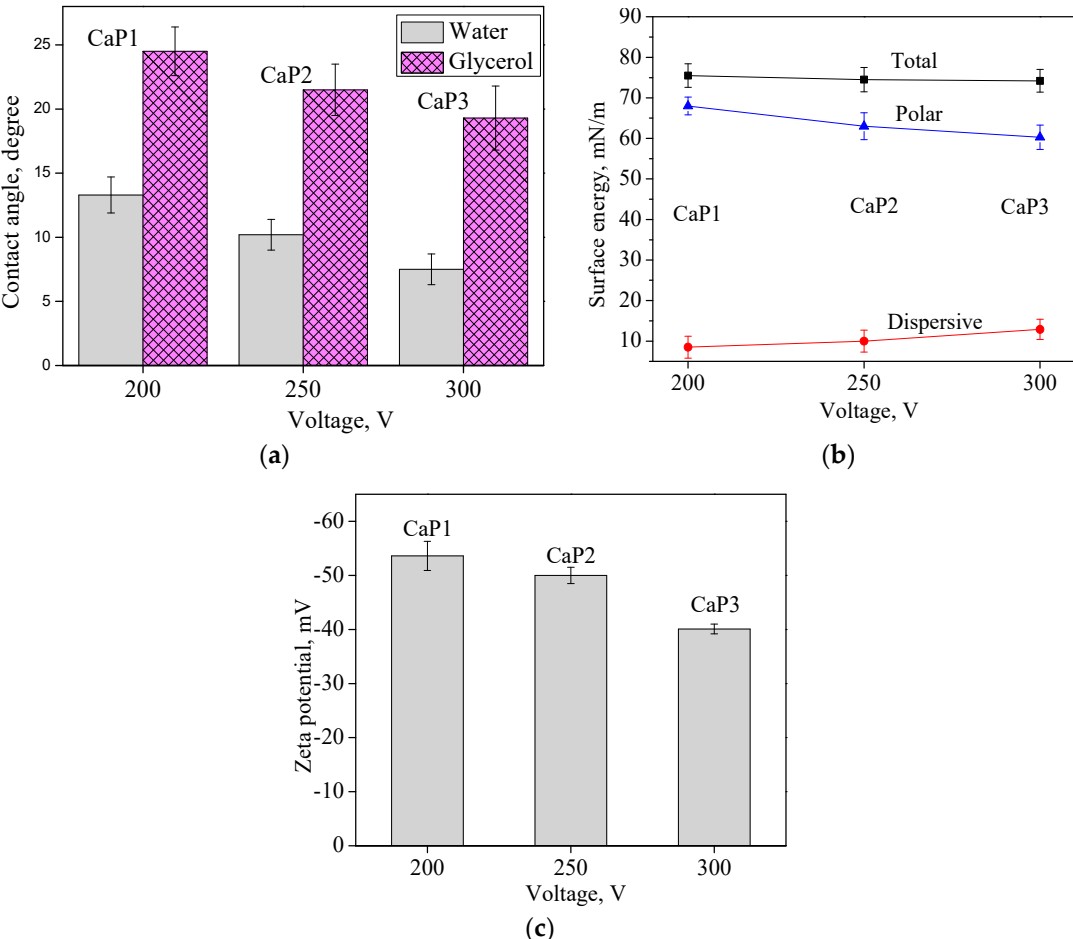

**Figure 7.** Dependences of contact angles (**a**), surface energy (**b**), and zeta potential (**c**) of CaP coatings on the MAO process voltage.

The research data showed that the zeta potential of CaP coatings has negative values (Figure 7c), which range from −40 to −53 mV. The negative charge of the coatings is associated with the mechanism of their formation during the MAO process in the anodic potentiostatic mode and in the electrolyte containing positive and negative ions. The MAO mechanism is a combination of a number of complex processes, one of which is the movement of charged ions to the counter electrode. The result of the process in a unipolar anode mode is the formation of calcium phosphate coatings with a predominantly negative charge. The coating, formed at a low voltage of 200 V, contained amorphous calcium phosphates and had a low Ca/P ratio of 0.3 and a minimum value of the zeta potential of −53 mV. As the MAO process voltage increased to 250–300 V, more powerful micro-arc discharges occurred. In this case, a larger amount of $Ca^{2+}$ ions were involved in the process; as a result, $CaHPO_4$ and $\beta$-$Ca_2P_2O_7$ crystalline compounds were formed in the coatings (Figure 6), so the Ca/P ratio increased up to 0.7. Herewith, the value of the zeta potential shifts to a less negative region and becomes −40 mV.

Thus, in the optimal voltage range of the MAO process from 200 to 300 V, porous CaP coatings on the surface of bioinert titanium were obtained. The main properties of the coatings depending on the applied voltage are presented in Table 1.

**Table 1.** Properties of CaP coatings depending on the applied voltage of the MAO process.

| Properties of the Coatings | Voltage of the MAO Process | | |
| --- | --- | --- | --- |
| | CaP1 −200 V | CaP2 −250 V | CaP3 −300 V |
| Pore size, μm | 4.0 ± 0.8 | 9.0 ± 1.0 | 15.1 ± 1.1 |
| Porosity, % | 31.5 ± 1.5 | 35.1 ± 2.0 | 37.5 ± 1.9 |
| Thickness, μm | 50 ± 2.5 | 80 ± 4.4 | 105 ± 5.3 |
| Roughness, $R_a$, μm | 2.9 ± 0.16 | 5.2 ± 0.49 | 6.8 ± 0.44 |
| Structure and the presence of crystalline compounds | Amorphous | Amorphous crystalline (monetite, calcium β-pyrophosphate) | Amorphous crystalline (monetite, calcium β-pyrophosphate) |
| Contact angle, water, ° | 13.3 ± 1.4 | 10.2 ± 1.2 | 7.5 ± 1.2 |
| Contact angle, glycerol, ° | 24.5 ± 1.9 | 21.5 ± 2 | 19.3 ± 2.5 |
| Surface energy, mN/m | 75.5 ± 2.9 | 74.5 ± 3 | 74.2 ± 2.8 |
| Zeta potential, mV | −53.6 ± 2.7 | −50.0 ± 1.5 | −40.1 ± 0.9 |

### 3.4. Kinetics of Adsorption-Desorption

The adsorption kinetics show the rate of the change in the concentration of the solution with time. To plot the adsorption kinetics, three groups of the coatings deposited at the different applied voltages (CaP1 −200 V, CaP2 −250 V, and CaP3 −300 V) were used. The main parameters of the coatings are given in Table 1. Based on the data obtained, kinetic adsorption curves were plotted, as shown in Figure 8a. The saturation of the DOX coating occurred within 60 min of keeping them in the solution. Then, the saturation ceased and the amount of the adsorbed DOX reached a plateau (Figure 8a). The maximum adsorption capacity was achieved after 150 min of the samples' exposure to the drug solution. The adsorption had a primarily physical nature and the drug chemical composition did not change in the process of impregnation. A comparative analysis of the adsorption curves allows concluding that, in accordance with the graphs, the amount of DOX adsorbed by the different types of coatings was changing as follows: CaP1 > CaP2 > CaP3. Although the CaP3 coating synthesized at a process voltage of 300 V had the highest porosity (37%) and pore size (15 μm), the amount of DOX adsorbed by the CaP3 for the same period was lower than that for the CaP1 and CaP2 coatings.

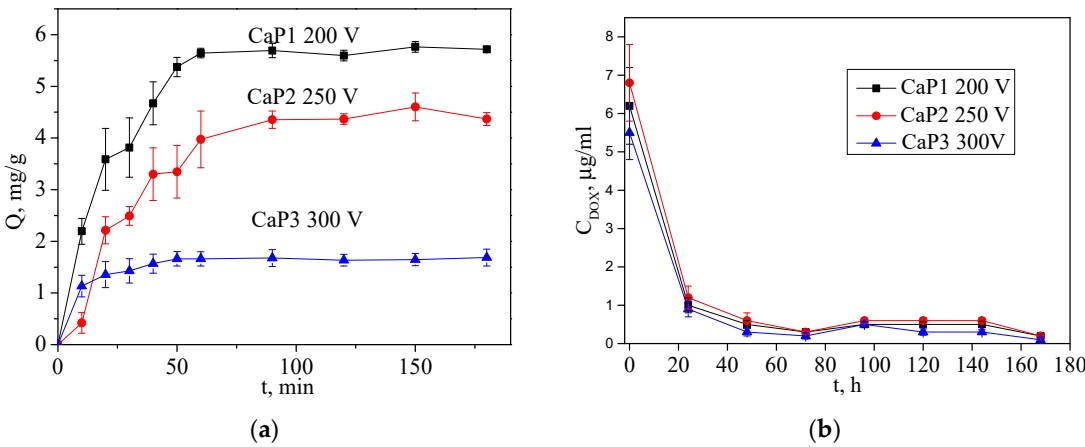

(**a**)　　　　　　　　　　　(**b**)

**Figure 8.** Kinetic curves of the adsorption (**a**) and desorption (**b**) of DOX by the samples of CaP coatings synthesized at different voltages of the MAO process.

This phenomenon can be explained as follows. At applied voltages of 200–250 V, the channel pores communicating with each other were formed in the coatings. When the voltage increased up to 300 V, the pores of a closed type were formed in the coating. The plates of the crystalline compounds $CaHPO_4$ and $β-Ca_2P_2O_7$ formed on the inner surface of pores can reduce their volume. Therefore, although the total porosity of CaP3 and its pore size reached its maximum, the volume of the substance that pores can adsorb was the smallest. In addition, although the surface energy for all types of the

coatings was close to 74 ± 3 mN/m, CaP1 had the highest negative zeta potential, indicative of its high sorption properties.

Figure 8b represents the desorption curves. As can be seen from the data obtained, the DOX desorption from the impregnated samples practically does not depend on the type of synthesized coating. The main process of DOX desorption from the coatings occurred on the first day of immersion of the impregnated coating samples in the solution. Then, all the analyzed concentrations of the released DOX were at the level of the detection limit of the method, which indicates its low elution from the samples. This can contribute to the low toxicity of the obtained samples and long-term action.

### 3.5. Effect of CaP Coatings Impregnated with DOX on the Viability of the Neuro-2a Cell Line

3.5.1. Study of the Effect of CaP Coatings Impregnated with DOX on the Viability of Neuro-2a Cell Line in the Goryaev Chamber

The samples of CaP coatings deposited at different voltages of the MAO process (Table 2) were selected to study the effect of DOX on the Neuro-2a cell line.

**Table 2.** Scheme of the CaP coatings treatment with DOX.

| Type of the Coating | Samples with DOX | Samples without DOX |
|---|---|---|
| CaP1 200 V | 1 | - |
| | 2 | - |
| | - | 3 |
| CaP2 250 V | 4 | - |
| | 5 | - |
| | - | 6 |
| CaP3 300 V | 7 | - |
| | 8 | - |
| | - | 9 |

After the contact of the CaP coating with the Neuro-2a cell culture, the culture medium did not change its color even after 24 h of incubation. The greatest decrease in the viability of Neuro-2a cells was observed (Figure 9) as a result of their contact with the CaP1 coating treated with DOX (6%–10% of viable cells). Figure 9 demonstrates cell morphology when incubated with DPBS (9a) and with the coating sample treated with DOX (9b). When the cells were incubated with DPBS, their morphology did not change. The cells in the division stage are clearly visible in the cell culture. After the incubation with the coating treated with DOX, a certain number of cells lost their typical shape and the cells sizes decreased significantly. The cells had pronounced morphological changes in the organelles and nuclear apparatus. In addition, there are elements of cells with damaged membranes.

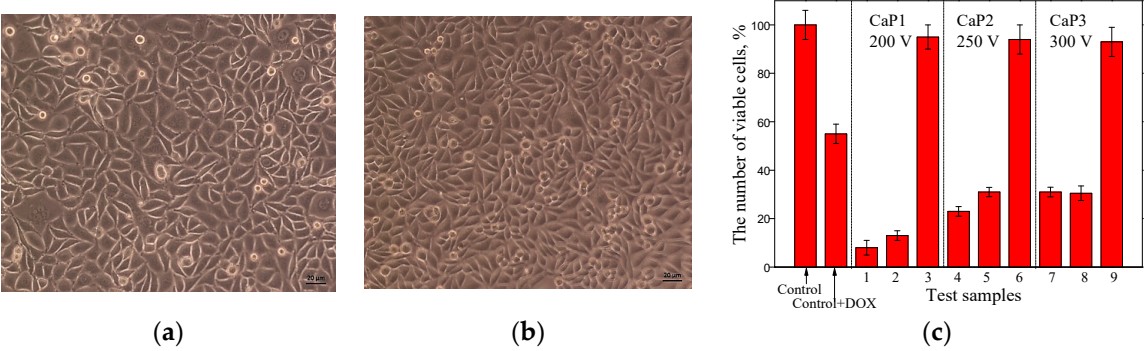

| (a) | (b) | (c) |

**Figure 9.** Photo of Neuro-2a cells in the control (**a**) and after incubation with a coating sample with DOX (**b**). Proliferation of tumor cells in the interaction with test samples of coatings (**c**).

The CaP2 and CaP3 coatings treated with DOX reduced the viability of cancer cells less intensely (22%–30% of viable cells). The coatings which were not treated with DOX did not significantly affect the cell line viability (Figure 9c).

### 3.5.2. Study of the Effect of CaP Coatings Impregnated with DOX on the Viability of Neuro-2a Cell Line by MTT Test

To study the effect of the CaP coatings impregnated with DOX on the viability of the Neuro-2a cell line by MTT test, the samples of CaP coatings obtained at different voltages of the MAO process and treated with DOX were prepared: CaP1 −200 V (10, 11, 12), CaP2 −250 V (13, 14, 15) and CaP3 −300 V (16, 17, 18). The dependences of the viability of the Neuro-2a cell line on the DOX desorption are presented in Figure 10. The highest cytotoxicity with respect to the cancer cells was observed for the CaP1 coating impregnated with DOX (50% of living cells in comparison with control). For the other types of coatings, the cytotoxic effect did not practically depend on the conditions of the CaP coatings' creation and the amount of DOX adsorbed. For all samples, the greatest cytotoxic effect was observed after the one-day immersion of the samples in the cell medium. Further studies have shown that the cytotoxicity decreases and remains almost constant throughout the experiment.

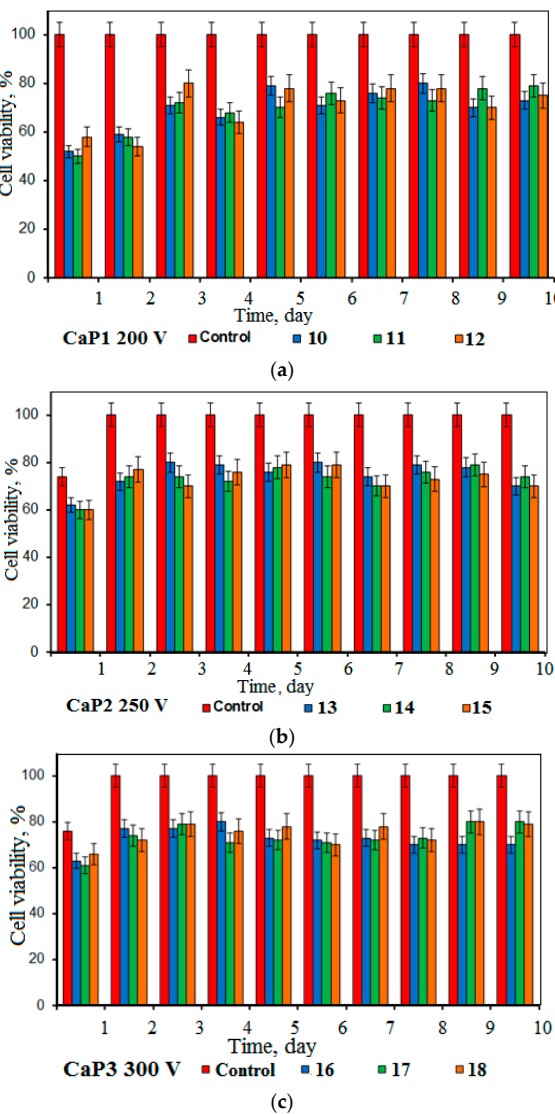

**Figure 10.** Results of the in vitro cytotoxic test of the CaP coating extracts for the Neuro-2a cell line compared with the control (100% viable cells). (**a**) CaP1; (**b**) CaP2; (**c**) CaP3.

## 4. Discussion

The samples of the CaP coatings formed by the MAO method on a Ti substrate at the voltages of 200, 250, and 300 V were impregnated with the chemotherapeutic drug DOX. Although the coatings CaP1 (200 V) and CaP2 (250 V) had a lower porosity and pore size than CaP3 (300 V), they were characterized by a higher sorption capacity.

Thus, it should be assumed that a variation in the thickness, roughness, and hydrophilic properties of the coatings does not significantly affect the amount of DOX that the CaP coating can adsorb. The roughness of the coatings was formed due to spheres, hemispheres, and pores formation, which was the results of micro-arc discharges. When the process voltage increased from 200 to 300 V, the intensity of the micro-arc discharges increased, and so the sizes of the spheres, hemispheres, and pores grew. As a result, the thickness and roughness of the coatings increased. It seems that the coating in this case should adsorb a larger amount of DOX [11]. However, the experiments showed the opposite result. This means that with a higher intensity of micro-arc discharges, the coatings melted and the internal pores were closed. In addition, during intense discharges the spheres were destroyed, and their fragments closed the outer pores. New crystalline phases were formed during the MAO process on the inner surface of the pores, closing the pore space. All these processes led to a decrease in the coatings' adsorption capacity. In this case, the porosity character had a greater effect. The coatings formed at low voltages of 200–250 V contained a greater number of branched communicating pores, and therefore they were able to adsorb a greater amount of DOX [12].

The adsorption process had a physical nature for the all the types of coatings because the chemical composition of the DOX did not change. The surface charge also contributes to the process of DOX adsorption, but this effect is not fully understood, and further studies are required to identify it. The authors [45] showed that the negative sign of the zeta potential of artificial material is preferable to stimulate osteogenesis. Negatively polarized surfaces can attract calcium ions, which accelerate the apatite nucleation [39]. $Ca^{2+}$ ions play a key role in osseointegration, since they stimulate the start of two processes: one is the nucleation and growth of HA; and the other is the adhesion of proteins such as integrin, fibronectin, and osteonectin [46]. In addition, polarized surfaces can also have an electrostatic effect on bacteria, since it is known that most bacteria and microorganisms have a negative charge. The authors [47] noted that the negatively charged surface of the biomaterial reduces the attachment of most bacteria and delays the development of biofilms.

The process of DOX desorption from the impregnated samples did not depend on the voltage of the MAO process at which the coatings were synthesized. The main process of DOX desorption from the samples occurred on the first day. Then, it took the drug a long time to desorb, which may be promising for the use of samples with CaP coatings as implants loaded with drugs for targeted delivery [1].

All the studied samples of CaP coatings containing DOX reduced the viability of Neuro-2a cells by more than 70%. The coatings without DOX did not significantly affect the viability of the cancer cell line. The greatest cytotoxic effect of DOX on the Neuro-2a cell line was observed after a day of exposure of the samples in the cell medium. Further investigation has shown that the cytotoxicity decreased and remained almost constant throughout the experiment.

## 5. Conclusions

CaP coatings were formed by the MAO method on the surface of titanium for the loading and controlled release of the anticancer drug DOX. As the applied voltage increased from 200 to 300 V, the coatings' thickness and roughness as well as the size of the spheres and pores increased. In this case, the wettability of the coatings rose, but the applied voltage did not affect the surface energy of the coatings. The negative value of the zeta potential was reduced. The high porosity of the coatings (above 30%), the presence of pores of up to 15 microns in size, the good hydrophilic properties (contact angle of wetting not higher than 25° ± 2°), the high surface energies of 75 ± 2 mN/m, and the negative zeta potential (−53 mV) indicate their ability to adsorb, retain, and effectively provide the controlled

release of drugs. The coating formed at low voltages of 200–250 V contained a greater number of branched communicating pores, and therefore they were able to adsorb a greater amount of DOX. The coating CaP1 deposited at the voltage 200 V and filled with DOX exhibited the greatest inhibition effect on the viability of the Neuro-2a cells (6%–10% viable cells when researched in the Goryaev chamber and 50% viable cells in the MTT assay).

**Author Contributions:** Conceptualization M.B.S., Y.P.S., and O.V.B.; data curation, M.B.S., V.V.C., E.G.K., and A.M.K.; formal analysis, M.B.S., V.V.C., and O.V.B.; investigation, E.G.K., T.V.T., V.V.C., A.M.K., and A.M.Z.; project administration, M.B.S. and Y.P.S.; supervision, M.B.S. and Y.P.S.; validation, M.B.S., O.V.B., and A.M.Z.; visualization, V.V.C. and A.M.K.; original draft, M.B.S.; review and editing of the final manuscript, M.B.S., Y.P.S., and O.V.B. All authors have read and agreed to the published version of the manuscript.

**Funding:** The work was financially supported by the Russian Federation via the Ministry of Science and Higher Education of the Russian Federation (Agreement No. 14.607.21.0202, project identifier RFMEFI60718X0202).

**Acknowledgments:** The authors express acknowledgement to A.I.T. and M.A.K. from the Institute of Strength Physics and Materials Science SB RAS (Tomsk. Russia) for their assistance in the preparation of the experimental materials.

**Conflicts of Interest:** The authors declare no conflict of interest.

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
