# Peer review of "Effect of the Porosity, Roughness, Wettability, and Charge of Micro-Arc Coatings on the Efficiency of Doxorubicin Delivery and Suppression of Cancer Cells"

_coatings, doi:10.3390/coatings10070664_

Round 1
Reviewer 1 Report
The authors presented an interesting paper, on MAO coatings, that can be impregnated with cytotoxic substances, and these effects could be used for cancer treatment.
Some small improvements could improve the quality of the text:
Line 113 a short description of the installation would make the reading of the text more accessible,
Line 128 does this method (XRD) has required accuracy? Is it possible that some other phases are formed during the MAO process, and that they are below the accuracy of the XRD
Line 167 physiological – please be more specific
Figure 4 – how the cross-section was prepared?
Figure 6 – general comment: for future investigations, authors could use TEM investigation for detailed phase analysis
Figure 8: what % of absorbed DOX was desorbed during the experiment?
Line 415 – how it is possible, that thickness does not affect the total amount of the DOX that can be absorbed? Is roughness not connected with internal porosity?
Line 432 should the desorption process, have a constant rate? Is it possible to control the desorption rate?
Reviewer 2 Report
In this manuscript, authors have demonstarted the effect of roughness, porosity, wettability, and the change of micro-arc coatings on the efficiency of doxorubicin delivery and suppression of cnacer cell. I have gone throughout the manuscript and found that this study is of a good interest and can be useful for future studies in this research area.
In my opinion, this manuscript can be accepted in its current form.
Author Response
Dear Reviewer, the Authors are very grateful to you for your kind review.
Reviewer 3 Report
Interesting paper. There are a few comments I would like to mention.
Minor changes
1 please spell out CaP, MAO and DOX, in the abstract
2 mm3 needs to be changed to mm3
3 you have spelled out doxorubicin or DOX randomly, please spell it out once in the introduction and then just day DOX
Major changes
1- why did you decide to culture cells and then put the samples on top to study cytotoxicity? the mechanical pressure of the sample on cells is totally ignored here. Did you use any culture well insert?
2 Material and methods are very lengthy while Discuttion is very short. The discussion has to be improved
Round 2
Reviewer 3 Report
Thank you for your answers. The manuscript has been improved